# Design Space for Graph Neural Networks

**Jiaxuan You**        **Rex Ying**        **Jure Leskovec**

Department of Computer Science, Stanford University
{jiaxuan, rexy, jure}@cs.stanford.edu

## Abstract

The rapid evolution of Graph Neural Networks (GNNs) has led to a growing number of new architectures as well as novel applications. However, current research focuses on proposing and evaluating specific *architectural designs* of GNNs, such as GCN, GIN, or GAT, as opposed to studying the more general *design space* of GNNs that consists of a Cartesian product of different *design dimensions*, such as the number of layers or the type of the aggregation function. Additionally, GNN designs are often specialized to a single task, yet few efforts have been made to understand how to quickly find the best GNN design for a novel task or a novel dataset. Here we define and systematically study the architectural design space for GNNs which consists of 315,000 different designs over 32 different predictive tasks. Our approach features three key innovations: (1) A general *GNN design space*; (2) a *GNN task space* with a *similarity metric*, so that for a given novel task/dataset, we can quickly identify/transfer the best performing architecture; (3) an efficient and effective *design space evaluation method* which allows insights to be distilled from a huge number of model-task combinations. Our key results include: (1) A comprehensive set of guidelines for designing well-performing GNNs; (2) while best GNN designs for different tasks vary significantly, the GNN task space allows for transferring the best designs across different tasks; (3) models discovered using our design space achieve state-of-the-art performance. Overall, our work offers a principled and scalable approach to transition from studying individual GNN designs for specific tasks, to systematically studying the GNN design space and the task space. Finally, we release GraphGym, a powerful platform for exploring different GNN designs and tasks. GraphGym features modularized GNN implementation, standardized GNN evaluation, and reproducible and scalable experiment management.

## 1 Introduction

The field of Graph Neural Network (GNN) research has made substantial progress in recent years. Notably, a growing number of GNN architectures, including GCN [16], GraphSAGE [6], and GAT [31], have been developed. These architectures are then applied to a growing number of applications, such as social networks [39, 46], chemistry [13, 44], and biology [52]. However, with this growing trend several issues emerge, which limit further development of GNNs.

**Issues in GNN architecture design**. In current GNN literature, GNN models are defined and evaluated as specific *architectural designs*. For example, architectures, such as GCN, GraphSAGE, GIN and GAT, are widely adopted in existing works [3, 4, 28, 41, 48, 37]. However, these models are specific instances in the GNN *design space* consisting of a cross product of *design dimensions*. For example, a GNN model that changes the aggregation function of GraphSAGE to a summation, or adds skip connections [8] across GraphSAGE layers, is not referred to as GraphSAGE model, but it could empirically outperform it in certain tasks [32]. Therefore, the practice of only focusing on a specific GNN *design*, rather than the *design space*, limits the discovery of successful GNN models.

**Issues in GNN evaluation**. GNN models are usually evaluated on a limited and non-diverse set of tasks, such as node classification on citation networks [15, 16, 31]. Recent efforts use additional

tasks to evaluate GNN models [3, 10]. However, while these new tasks enrich the evaluation of GNN models, challenging and completely new tasks from various domains always emerge. For example, novel tasks such as circuit design [50], SAT generation [47], data imputation [45], or subgraph matching [40] have all been recently approached with GNNs. Such novel tasks do not naturally resemble any of the existing GNN benchmark tasks and thus, it is unclear how to design an effective GNN architecture for a given new task. This issue is especially critical considering the large design space of GNNs and a surge of new GNN tasks, since re-exploring the entire design space for each new task is prohibitively expensive.

**Issues in GNN implementation**. A platform that supports extensive exploration over the GNN design space, with unifies implementation for node, edge, and graph-level tasks, is currently lacking, which is a major factor that contributes to the above-mentioned issues.

**Present work**. Here we develop and systematically study a general design space of GNNs over a number of diverse of tasks[1]. To tackle the above-mentioned issues, we highlight three central components in our study, namely *GNN design space*, *GNN task space*, and *design space evaluation*: **(1)** The GNN design space covers important architectural design aspects that researchers often encounter during GNN model development. **(2)** The GNN task space with a task similarity metric allows us to identify novel tasks and effectively transfer GNN architectural designs between similar tasks. **(3)** An efficient and effective design space evaluation allows insights to be distilled from a huge number of model-task combinations. Finally, we also develop GraphGym[2], a platform for exploring different GNN designs and tasks, which features modularized GNN implementation, standardized GNN evaluation, and reproducible and scalable experiment management.

**GNN design space**. We define a general design space of GNNs that considers *intra-layer design*, *inter-layer design* and *learning configuration*. The design space consists of 12 design dimensions, resulting in 315,000 possible designs. Our purpose is thus not to propose the most extensive GNN design space, but to demonstrate how focusing on the design space can enhance GNN research. We emphasize that the design space can be expanded as new design dimensions emerge in state-of-the-art models. Our overall framework is easily extendable to new design dimensions. Furthermore, it can be used to quickly find a good combination of design choices for a specific novel task.

**GNN task space**. We propose a *task similarity metric* to characterize relationships between different tasks. The metric allows us to quickly identify promising GNN designs for a brand new task/dataset. Specifically, the similarity between two tasks is computed by applying a fixed set of GNN architectures to the two tasks and then measuring the Kendall rank correlation [1] of the performance of these GNNs. We consider 32 tasks consisting of 12 synthetic node classification tasks, 8 synthetic graph classification tasks, and 6 real-world node classification and 6 graph classification tasks.

**Design space evaluation**. Our goal is to gain insights from the defined GNN design space, such as *"Is batch normalization generally useful for GNNs?"* However, the defined design and task space lead to over 10M possible combinations, prohibiting a full grid search. Thus, we develop a *controlled random search* evaluation procedure to efficiently understand the trade-offs of each design dimension.

Based on these innovations, our work provides the following key results: **(1)** A comprehensive set of guidelines for designing well-performing GNNs (Sec. 7.3). **(2)** While best GNN designs for different tasks/datasets vary significantly, the defined GNN task space allows for transferring the best designs across tasks (Sec. 7.4). This saves redundant algorithm development efforts for highly similar GNN tasks, while also being able to identify novel GNN tasks which can inspire new GNN designs. **(3)** Models discovered from our design space achieve state-of-the-art performance on a new task from the Open Graph Benchmark [10] (Sec. 7.5).

Overall, our work suggests a transition from studying specific GNN designs for specific tasks to studying the GNN design space, which offers exciting opportunities for GNN architecture design. Our work also facilitates reproducible GNN research, where GNN models and tasks are precisely described and then evaluated using a standardized protocol. Using the proposed GraphGym platform reproducing experiments and fairly comparing models requires minimal effort.

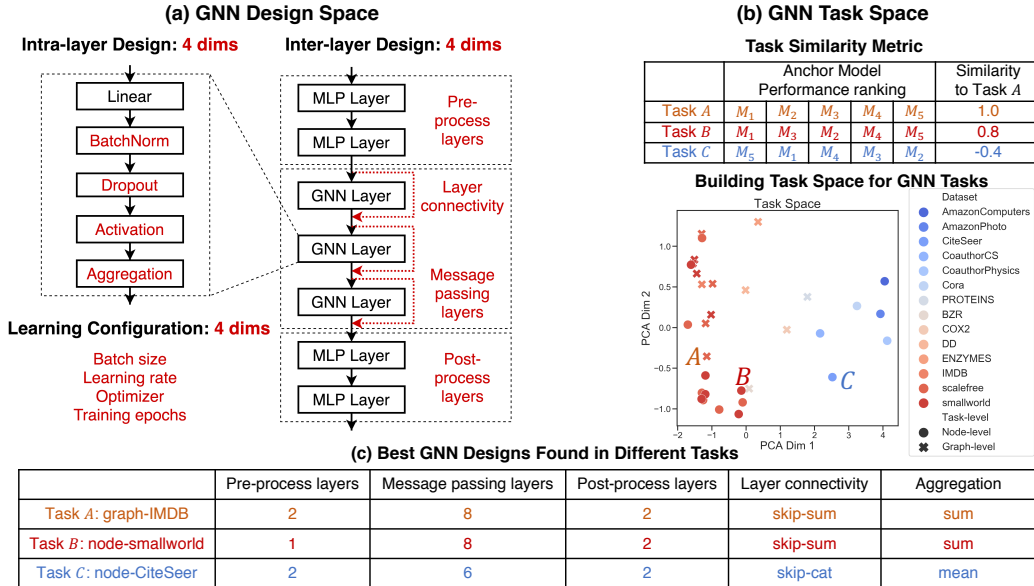

**(a) GNN Design Space**

**Intra-layer Design: 4 dims**　　**Inter-layer Design: 4 dims**

Linear → BatchNorm → Dropout → Activation → Aggregation

MLP Layer → MLP Layer　Pre-process layers
GNN Layer　Layer connectivity
GNN Layer
GNN Layer　Message passing layers
MLP Layer → MLP Layer　Post-process layers

**Learning Configuration: 4 dims**

Batch size
Learning rate
Optimizer
Training epochs

**(b) GNN Task Space**

**Task Similarity Metric**

| | Anchor Model Performance ranking | | | | | Similarity to Task $A$ |
|---|---|---|---|---|---|---|
| Task $A$ | $M_1$ | $M_2$ | $M_3$ | $M_4$ | $M_5$ | 1.0 |
| Task $B$ | $M_1$ | $M_3$ | $M_2$ | $M_4$ | $M_5$ | 0.8 |
| Task $C$ | $M_5$ | $M_1$ | $M_4$ | $M_3$ | $M_2$ | -0.4 |

**Building Task Space for GNN Tasks**

**(c) Best GNN Designs Found in Different Tasks**

| | Pre-process layers | Message passing layers | Post-process layers | Layer connectivity | Aggregation |
|---|---|---|---|---|---|
| Task $A$: graph-IMDB | 2 | 8 | 2 | skip-sum | sum |
| Task $B$: node-smallworld | 1 | 8 | 2 | skip-sum | sum |
| Task $C$: node-CiteSeer | 2 | 6 | 2 | skip-cat | mean |

Figure 1: **Overview of the proposed GNN design and task space**. **(a)** A GNN design space consists of 12 design dimensions for intra-layer design, inter-layer design and learning configuration. **(b)** We apply a fixed set of "anchor models" to different tasks/datastes, then use the Kendall rank correlation of their performance to quantify the similarity between different tasks. This way, we build the GNN task space with a proper similarity metric. **(c)** The best GNN designs for tasks $A$, $B$, $C$. Notice that tasks with higher similarity share similar designs, indicating the efficacy of our GNN task space.

## 2 Related Work

**Graph Architecture Search**. Architecture search techniques have been applied to GNNs [5, 51]. However, these works only focus on the design within each GNN layer instead of a general GNN design space, and only evaluate the designs on a small number of node classification tasks.

**Evaluation of GNN Models**. Multiple works discuss approaches for making fair comparison between GNN models [3, 4, 28]. However, these models only consider some specific GNN designs (*e.g.*, GCN, GAT, GraphSAGE), while our approach extensively explores the general design space of GNNs.

**Other graph learning models**. We focus on message passing GNNs due to their proven performance and efficient implementation over various GNN tasks. There are alternative designs of graph learning models [18–20, 48], but their design spaces are different from GNNs and are less modularized.

**Transferable Architecture Search**. The idea of transferring architecture search results across tasks has been studied in the context of computer vision tasks [43, 53]. Meta-level architecture design has also been studied in [23, 27, 36, 49], with the assumption that different tasks follow the same distribution (*e.g.*, variants of ImageNet dataset [2]). These approaches often make an assumption that a single neural architecture may perform well on all tasks, which fits well for tasks with relatively low variety. However, due to the great variety of graph learning tasks, such assumption no longer holds.

## 3 Preliminaries

Here we outline the terminology used in this paper. We use the term *design* to refer to a concrete GNN instantiation, such as a 5-layer GraphSAGE. Each design can be characterized by multiple *design dimensions* such as the number of layers $L \in \{2, 4, 6, 8\}$ or the type of aggregation function $\text{AGG} \in \{\text{MAX}, \text{MEAN}, \text{SUM}\}$, where a *design choice* is the actual selected value in the design dimension, such as $L = 2$. A *design space* consists of a Cartesian product of design dimensions. For example, a design space with design dimensions $L$ and $\text{AGG}$ has $4 \times 3 = 12$ possible designs. A GNN can be applied to a variety of *tasks*, such as node classification on Cora [26] dataset or graph classification on ENZYMES dataset [14], constituting a *task space*. Applying a GNN design to a task is referred to as an *experiment*. An *experiment space* covers all combinations of designs and tasks.

# 4 Proposed Design Space for GNNs

Next we propose a general design space for GNNs, which includes three crucial aspects of GNN architecture design: *intra-layer design*, *inter-layer design* and *learning configuration*. We use the following principles when defining the design space: **(1)** covering most important design dimensions that researchers encounter during model development; **(2)** including as few design dimensions as possible (*e.g.*, we drop model-specific design dimensions such as dimensions for attention modules); **(3)** considering modest ranges of options in each design dimension, based on reviewing a large body of literature and our own experience. Our purpose is not to propose the most extensive design space, but to demonstrate how focusing on the design space can help inform GNN research.

**Intra-layer design**. GNN consists of several message passing layers, where each layer could have diverse design dimensions. As illustrated in Figure 1(a), the adopted GNN layer has a linear layer, followed by a sequence of modules: **(1)** batch normalization $\text{BN}(\cdot)$ [12]; **(2)** dropout $\text{DROPOUT}(\cdot)$ [29]; **(3)** nonlinear activation function $\text{ACT}(\cdot)$, where we consider $\text{RELU}(\cdot)$ [21], $\text{PRELU}(\cdot)$ [7] and $\text{SWISH}(\cdot)$ [24]; **(4)** aggregation function $\text{AGG}(\cdot)$. Formally, the $k$-th GNN layer can be defined as:

$$\mathbf{h}_v^{(k+1)} = \text{AGG}\Big(\Big\{\text{ACT}\Big(\text{DROPOUT}\big(\text{BN}(\mathbf{W}^{(k)}\mathbf{h}_u^{(k)} + \mathbf{b}^{(k)})\big)\Big), u \in \mathcal{N}(v)\Big\}\Big)$$

where $\mathbf{h}_v^{(k)}$ is the $k$-th layer embedding of node $v$, $\mathbf{W}^{(k)}, \mathbf{b}^{(k)}$ are trainable weights, and $\mathcal{N}(v)$ is the local neighborhood of $v$. We consider the following ranges for the design dimensions:

| Batch Normalization | Dropout | Activation | Aggregation |
|---|---|---|---|
| True, False | False, 0.3, 0.6 | RELU, PRELU, SWISH | MEAN, MAX, SUM |

**Inter-layer design**. Having defined a GNN layer, another level of design is how these layers are organized into a neural network. In GNN literature, the common practice is to directly stack multiple GNN layers [16, 31]. Skip connections have been used in some GNN models [6, 17, 38], but have not been systematically explored in combination with other design dimensions. Here we investigate two choices of skip connections: residual connections SKIP-SUM [8], and dense connections SKIP-CAT that concatenate embeddings in all previous layers [11]. We further explore adding Multilayer Perceptron (MLP) layers before/after GNN message passing. All these design alternatives have been shown to benefit performance in some cases [42, 44]. In summary, we consider these design dimensions:

| Layer connectivity | Pre-process layers | Message passing layers | Post-precess layers |
|---|---|---|---|
| STACK, SKIP-SUM, SKIP-CAT | 1, 2, 3 | 2, 4, 6, 8 | 1, 2, 3 |

**Training configurations**. Optimization algorithm plays an important role in GNN performance. In GNN literature, training configurations including batch size, learning rate, optimizer type and training epochs often vary a lot. Here we consider the following design dimensions for GNN training:

| Batch size | Learning rate | Optimizer | Training epochs |
|---|---|---|---|
| 16, 32, 64 | 0.1, 0.01, 0.001 | SGD, ADAM | 100, 200, 400 |

# 5 Proposed Task Space for GNNs

One of our key insights is that the design space of GNN should be studied *in conjunction with the task space*, because different tasks may have very different best-performing GNN designs (Figure 1(c)). Here, explicitly creating the task space is challenging because researchers can apply GNNs to an ever increasing set of diverse tasks and datasets. Thus, we develop techniques to *measure* and *quantify* the GNN task space, rather than restricting ourselves to a fixed taxonomy of GNN tasks. To validate our approach, we collect 32 diverse GNN tasks as illustrative examples, but our approach is general and can be applied to any other novel GNN task.

## 5.1 Quantitative Task Similarity Metric

**Issues in existing task taxonomy**. GNN tasks have been categorized either by dataset domains such as biological or social networks, or by the prediction types such as node or graph classification. However, these taxonomies do not necessarily imply transferability of GNN designs between tasks/datasets. For example, two tasks may both belong to node classification over social networks, but different types of node features could result in different GNN designs performing best.

**Proposed task similarity metric**. We propose to quantitatively measure the similarity between tasks, which is crucial for **(1)** transferring best GNN designs or design principles across tasks that are similar, and **(2)** identifying novel GNN tasks that are not similar to any existing task, which can inspire new GNN designs. The proposed task similarity metric consists of two components: **(1)** selection of *anchor models* and **(2)** measuring the *rank distance* of the performance of anchor models.

**Selection of anchor models**. Our goal is to find the most diverse set of GNN designs that can reveal different aspects of a given GNN task. Specifically, we first sample $D$ random GNN designs from the design space. Then, we apply these designs to a fixed set of GNN tasks, and record each GNN's average performance across the tasks. The $D$ designs are ranked and evenly sliced into $M$ groups, where models with median performance in each group are selected. Together, these $M$ selected GNN designs constitute the set of anchor models, which are fixed for all further task similarity computation.

**Measure task similarity**. Given two distinct tasks, we first apply all $M$ anchor models to these tasks and record their performance. We then rank the performance of all $M$ anchor models for each of the tasks. Finally, we use Kendall rank correlation [1] to compute the similarity score between the rankings of the $M$ anchor models, which is reported as task similarity. When $T$ tasks are considered, rank distances are computed for all pairs of tasks. Overall, the computation cost to compare $T$ GNN tasks is to train and evaluate $M * T$ GNN models. We show that $M = 12$ anchor models is sufficient to approximate the task similarity computed with all the designs in the design space (Figure 5(b)).

Our approach is *fully general* as it can be applied to any set of GNN tasks. For example, in binary classification ROC AUC score can be used to rank the models, while in regression tasks mean square error could be used. Our task similarity metric can even generalize to non-predictive tasks, *e.g.*, molecule generation [44], where the drug-likeness score of the generated molecular graphs can be used to rank different GNN models.

### 5.2 Collecting Diverse GNN Tasks

To properly evaluate the proposed design space and the proposed task similarity metric, we collect a variety of 32 synthetic and real-world GNN tasks/datasets. Our overall principle is to select medium-sized, diverse and realistic tasks, so that the exploration of GNN design space can be efficiently conducted. We focus on node and graph level tasks; results for link prediction are in the Appendix.

**Synthetic tasks**. We aim to generate synthetic tasks with diverse graph *structural properties*, *features*, and *labels*. We use two families of graphs prevalent in the real-world small-world [33] and scale-free graphs [9], which have diverse structural properties, measured by a suite of *graph statistics*. We consider a local graph statistic Average Clustering Coefficient $C$, and a global graph statistic, Average Path Length $L$. We generate graphs to cover the ranges $C \in [0.3, 0.6]$ and $L \in [1.8, 3.0]$. We create an 8x8 grid over these two ranges, and keep generating each type of random graphs until each bin in the grids have 4 graphs. In all, we have 256 small-world graphs and 256 scale-free graphs.

We consider four types of node features: **(1)** constant scalar, **(2)** one-hot vectors, **(3)** node clustering coefficients and **(4)** node PageRank score [22]. We consider node-level labels including node clustering coefficient and node PageRank score, and graph-level labels, such as average path length. We exclude the combination where features and labels are the same. We perform binning over these continuous labels into 10 bins and perform 10-way classification. Altogether, we have 12 node classification tasks and 8 graph classification tasks. All the tasks are listed as axes in Figure 5(a).

**Real-world tasks**. We include 6 node classification benchmarks from [26, 28], and 6 graph classification tasks from [14]. The tasks are also listed as axes in Figure 5(a).

## 6 Evaluation of GNN Design Space

The defined design space and the task space lead to over 10M possible combinations, prohibiting the full grid search. To overcome this issue, we propose an efficient and effective design space evaluation technique, which allows insights to be distilled from the huge number of model-task combinations. As summarized in Figure 2, our approach is based on a novel *controlled random search* technique. Here the computational budgets for all the models are controlled to ensure a fair comparison.

**Controlled random search**. Figure 2 illustrates our approach. Suppose we want to study whether adding BatchNorm (BN) is generally helpful for GNNs. We first draw $S$ random experiments from

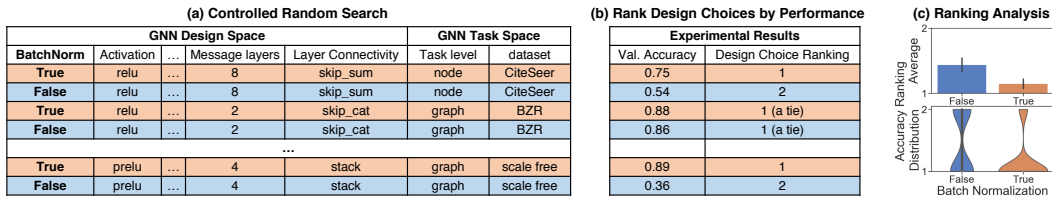

Figure 2: **Overview of the proposed evaluation of GNN design space**. **(a)** A controlled random search technique is used to study the effect of BatchNorm. **(b)** We rank the design choice of BN = TRUE and FALSE by their performance in each setup, where a tie is reached if the performances are close. **(c)** Insights on different design choices can be obtained from the distribution of the rankings.

the 10M possible model-task combinations, all with BN = TRUE. We set $S = 96$ so that each task has 3 hits on average. We then alter these 96 setups to have BN = FALSE while *controlling all the other dimensions* (Figure 2(a)). We rank the design choices of BN $\in$ [TRUE, FALSE] within each of the 96 setups by their performance (Figure 2(b)). To improve the robustness of the ranking results, we consider the case of a tie, if the performance difference falls within $\epsilon = 0.02$. Finally, we collect the ranking over the 96 setups, and analyze the distribution of the rankings (Figure 2(c)). In our experiment, BN = TRUE has an average rank of 1.15, while the average rank of BN = FALSE is 1.44, indicating that adding BatchNorm to GNNs is generally helpful. This controlled random search technique can be easily generalized to design dimensions with multiple design choices. Our approach cuts the number of experiments by over 1,000 times compared with a full grid search: a full evaluation of all 12 design choices over 32 tasks only takes about 5 hours on 10 GPUs.

**Controlling the computational budget**. To ensure fair comparisons, we additionally control the number of trainable parameters of GNNs for all experiments in our evaluation. Specifically, we use a GNN with a stack of 1 pre-processing layer, 3 message passing layers, 1 post-processing layer and 256 hidden dimensions to set the computational budget. For all the other GNN designs in the experiments, the number of hidden dimensions is adjusted to match this computational budget.

# 7   Experiments

## 7.1   `GraphGym`: Platform for GNN Design

We design `GraphGym`, a novel platform for exploring GNN designs. We believe `GraphGym` can greatly facilitate the research field of GNNs. Its highlights are summarized below.

**Modularized GNN implementation**. `GraphGym`'s implementation closely follows the proposed general GNN design space. Additionally, users can easily import new design dimensions to `GraphGym`, such as new types of GNN layers or new connectivity patterns across layers. We provide an example of using ATTENTION as a new intra-layer design dimension in the Appendix.

**Standardized GNN evaluation**. `GraphGym` provides a standarized evaluation pipeline for GNN models. Users can select how to split the dataset (*e.g.*, customized split or random split, train/val split or train/val/test split), what metrics to use (*e.g.*, accuracy, ROC AUC, F1 score), and how to report the performance (*e.g.*, final epoch or the best validation epoch).

**Reproducible and scalable experiment management**. In `GraphGym`, any experiment is precisely described by a configuration file, so that results can be *reliably reproduced*. Consequently, **(1)** sharing and comparing novel GNN models require minimal effort; **(2)** algorithmic advancement of the research field can be easily tracked; **(3)** researchers outside the community can get familiar with advanced GNN designs at ease. Moreover, `GraphGym` has full support for parallel launching, gathering and analyzing *thousands of experiments*. The analyses in this paper can be automatically generated by running just a few lines of code.

## 7.2   Experimental Setup

We evaluate the proposed GNN design space (Section 4) over the GNN task space (Section 5), using the proposed evaluation techniques (Section 6). For all the experiments in Sections 7.3 and 7.4, we use a consistent setup, where results on three random 80%/20% train/val splits are averaged, and the validation performance in the final epoch is reported. Accuracy is used for multi-way classification and ROC AUC is used for binary classification. For graph classification tasks the splits are always

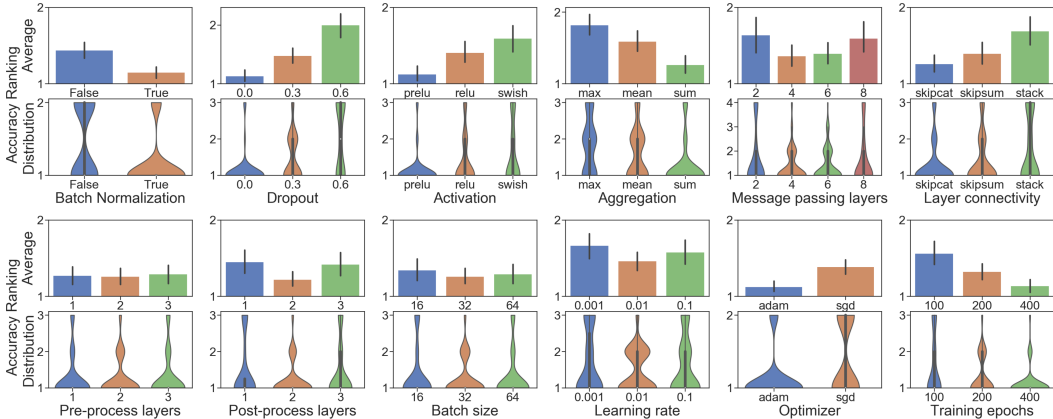

Figure 3: **Ranking analysis for GNN design choices in all 12 design dimensions**. Lower is better. A tie is reached if designs have accuracy / ROC AUC differences within $\epsilon = 0.02$.

inductive (tested on unseen graphs); for node classification tasks over multi-graph datasets, the split can either be inductive or transductive (tested on unseen nodes on a training graph), where we select the transductive setting to diversify the tasks. More details are provided in the Appendix.

### 7.3 Results on GNN Design Space Evaluation

**Evaluation with ranking analysis**. We apply the evaluation technique from Section 6 to each of the 12 design dimensions of the proposed design space, where 96 experimental setups are sampled for each design dimension. Results are summarized in Figure 3, where in each subplot, rankings of each design choice are aggregated over all the 96 setups via bar plot and violin plot. The bar plot shows the average ranking across all the 96 setups (lower is better). An average ranking of 1 indicates the design choice always achieves the best performance among other choices of this design dimension, for all the 96 sampled setups. The violin plot indicates the smoothed distribution of the ranking of each design choice over all the 96 setups. Given that tied 1st rankings commonly exist, violin plots are very helpful for understanding how often a design choice is *not* being ranked 1st.

**Findings on intra-layer design**. Figure 3 shows several interesting findings for intra-layer GNN design. **(1)** Adding BN is generally helpful. The results confirm previous findings in general neural architectures that BN facilitates neural network training [12, 25]. **(2)** For GNNs, dropping out node feature dimensions as a mean of regularization is usually not effective. We think that this is because GNNs already involve neighborhood aggregation and are thus robust to noise and outliers. **(3)** PRELU clearly stands out as the choice of activation. We highlight the novelty of this discovery, as PRELU has rarely been used in existing GNN designs. **(4)** SUM aggregation is theoretically most expressive [37], which aligns with our findings. In fact, we provide the first comprehensive and rigorous evaluation that verifies SUM is indeed empirically successful.

**Findings on inter-layer design**. Figure 3 further shows findings for inter-layer design. **(1)** There is no definitive conclusion for the best number of message passing layers. Take choices of 2 and 8 message passing layers, which perform best in very different tasks. **(2)** Adding skip connections is generally favorable, and the concatenation version (SKIP-CAT) results in a slightly better performance than SKIP-SUM. **(3)** Similarly to (1), the conclusions for the number of pre-processing layers and post-processing layers are also task specific.

**Findings on learning configurations**. Figure 3 also shows findings for learning configurations. **(1)** Batch size of 32 is a safer choice, as it has significantly lower probability mass of being ranked 3rd. **(2)** Learning rate of 0.01 is also favorable, also because it has significantly lower probability mass of being ranked 3rd. **(3)** ADAM is generally better than a naive SGD, although it is known that tuned SGD can have better performance [35]. **(4)** More epochs of training lead to better performance.

Overall, the proposed evaluation framework provides a solid tool to *rigorously verify GNN design dimensions*. By conducting controlled random search over 10M possible model-task combinations (96 experiments per design dimension), our approach provides a *more convincing guideline on GNN designs*, compared with the common practice that only evaluate a new design dimension on a fixed GNN design (*e.g.*, 5-layer, 64-dim, etc.) on a few graph prediction tasks (*e.g.*, Cora or ENZYMES).

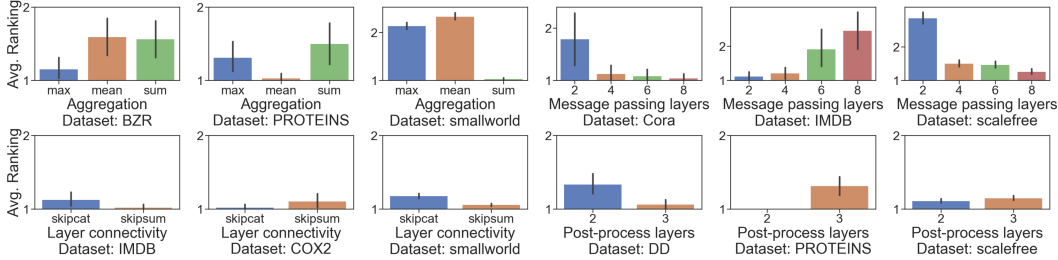

Figure 4: **Ranking analysis for GNN design choices over different GNN tasks**. Lower is better. Preferable design choices greatly vary across GNN tasks.

Additionally, we verify that our findings do not suffer from the problem of multiple hypothesis testing: we find that 7 out of the 12 design dimensions have a significant influence on the GNN performance, under one-way ANOVA [30] with Bonferroni correction [34] (p-value 0.05).

### 7.4 Results on the Efficacy of GNN Task Space

**Condensed GNN design space**. Based on the guidelines we discovered in Section 7.3, we fix several design dimensions to condense the GNN design space. A condensed design space enables as to perform a full grid search, which greatly helps at: **(1)** verifying if the proposed task space is indeed informative for transferring the best GNN design across GNN tasks; **(2)** applying our approach to larger-scale datasets to verify if it provides empirical benefits (Section 7.5). Specifically, we fix a subset of design choices as shown in Table 7.3.

Table 1: Condensed GNN design space based on the analysis in Section 7.3

| Activation | BN | Dropout | Aggregation | MP layers | Pre-MP layers | Post-MP layers | Connectivity | Batch | LR | Optimizer | Epoch |
|---|---|---|---|---|---|---|---|---|---|---|---|
| PRELU | True | False | MEAN, MAX, SUM | 2,4,6,8 | 1,2 | 2,3 | SKIP-SUM, SKIP-CAT | 32 | 0.01 | ADAM | 400 |

**Preferable design choices vary greatly across tasks**. Following the approach in Section 7.3, we use bar plots to demonstrate how the preferable GNN designs can significantly differ across different tasks/datasets. As is shown in Figure 4, the findings are surprising: desirable design choices for Aggregation, Message passing layers, layer connectivity and post-processing layers drastically vary across tasks. We show the specification of the best GNN design for each of the tasks in the Appendix.

**Building task space via the proposed task similarity metric**. To further understand when preferable designs can transfer across tasks, we follow the technique in Section 5 to build a GNN task space for all 32 tasks considered in this paper. Figure 5(a) visualizes the task similarities computed using the proposed metric. The key finding is that tasks can be roughly clustered into two groups: **(1)** node classification over real-world graphs, **(2)** node classification over synthetic graphs and all graph classification tasks. Our understanding is that tasks in (1) are node-level tasks with rich node features, thus GNN designs that can better propagate *feature information* are preferred; in contrast, tasks in (2) require graph *structural information* and thus call for different GNN designs.

**Best GNN design can transfer to tasks with high similarity**. We transfer the best model in one task to another task, then compute the performance ranking of this model in the new task. We observe a high 0.8 Pearson correlation between the performance ranking after task transfer and the similarity of the two tasks in Figure 5(c), which implies that the proposed task similarity metric can indicate how well a GNN design transfers to a new task. Note that this finding implies that finding a good model in a new task can be highly efficient: computing task similarity only requires running 12 models, as opposed to a grid search over the full design space (315,000 designs).

**Comparing best designs with standard GNN designs**. To further demonstrate the efficacy of our design space, we further compare the best designs in the condensed design space with standard GNN designs. We implement standard GCNs with message passing layers $\{2, 4, 6, 8\}$, while *using all the other optimal hyper-parameters* that we have discovered in Table 1. The best model in our design space is better than the best GCN model in 24 out of 32 tasks. Concrete performance comparisons are shown in the Appendix. We emphasize that the goal of our paper is not to pursue SOTA performance, but to present a systematic approach for GNN design.

### 7.5 Case Study: Applying to a Challenging New Task `ogbg-molhiv`

**Background**. Our insights on GNN design space (Section 4) and task space (Section 5) can lead to empirically successful GNN designs on challenging new tasks. Recall that from the proposed

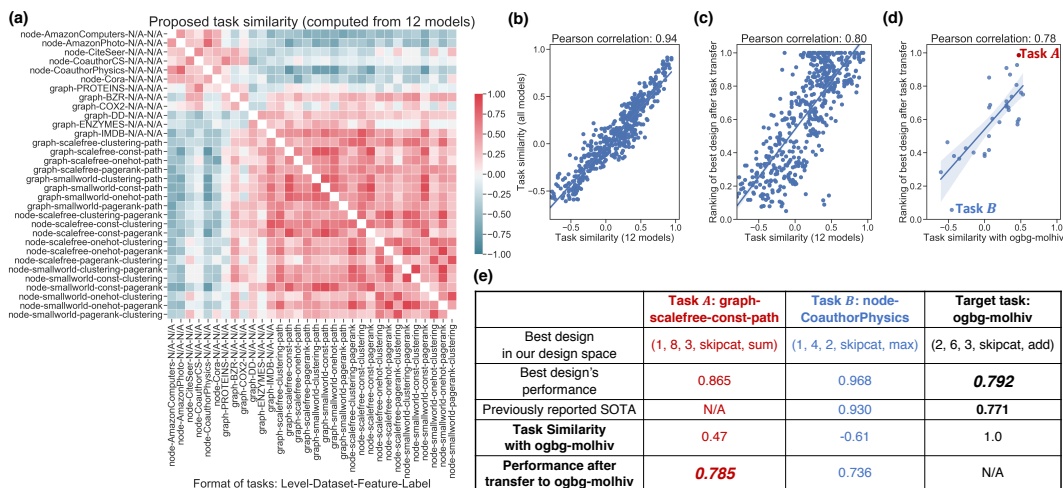

Figure 5: **(a)** Proposed similarity between all pairs of tasks computed using 12 anchor models, higher value means higher similarity. We collect all pairs of tasks, and show: **(b)** the correlation between task similarity computed using 12 anchor models versus using all 96 models; **(c)** the correlation between task similarity versus the performance ranking of the best design in one task after transferred to another task; **(d)** the correlation between task similarity with `ogbg-molhiv` versus the performance ranking of the best model in one task after transferred to `ogbg-molhiv`; **(e)** the best design found in a task with high measured similarity with `ogbg-molhiv` can achieves state-of-the-art performance.

GNN design space, we obtain a useful guideline (Section 7.3) that condenses the design space from 315,000 designs to just 96 designs; from the proposed GNN task space, we build a task similarity metric where top GNN designs can transfer to tasks with high measured similarity (Section 7.4).

**Setup**. Specifically, we use `ogbg-molhiv` dataset [10], which appears to be very different from the 32 tasks we have investigated: it is much larger (41K graphs *vs.* maximum 2K graphs), highly imbalanced (1.4% positive labels) and requires out-of-distribution generalization (split by molecule structure versus split at random). We use the train/val/test splits provided by the dataset, and report the test accuracy in the final epoch. We match the model complexity of current state-of-the-art (SOTA) model (with ROC AUC 0.771) [10], and follow their practice of training 100 epochs due to the high computational cost; except for these, we follow *all* the design choices in the condensed GNN design space (Section 7.3). We examine: **(1)** if the best model in the condensed design space can achieve SOTA performance, **(2)** if the task similarity can guide transferring top designs *without grid search*.

**Results**. Regarding (1), in Figure 5(e) last column, we show that the best GNN discovered in our condensed design space significantly outperforms existing SOTA (ROC AUC 0.792 versus 0.771). As for (2), we first verify in Figure 5(d) that the proposed task similarity remains to be a reasonable indicator of how well a top GNN design can transfer across tasks, and that out of the 32 tasks we have investigated, we select tasks $A$ and $B$, which have 0.47 and -0.61 similarity to `ogbg-molhiv` respectively. We show in Figure 5(e) that although the best design for task $B$ significantly outperforms existing SOTA (accuracy 0.968 versus 0.930), after transferred to `ogbg-molhiv` it performs poorly (AUC 0.736). In contrast, since task $A$ has a high measured similarity, *directly* using the best design in task $A$ already significantly outperforms SOTA on `ogbg-molhiv` (ROC AUC 0.785 versus 0.771).

## 8   Conclusion

In this paper we offered a principled approach to building a general *GNN design space* and a *GNN task space* with quantitative similarity metric. Our extensive experimental results showed that coherently studying both spaces via tractable *design space evaluation* techniques can lead to exciting new understandings of GNN models and tasks, saving algorithm development costs as well as empirical performance gains. Overall, our work suggests a transition from studying individual GNN designs and tasks to systematically studying a GNN design space and a GNN task space.

## Broader Impact

**Impact on GNN research**. Our work brings in many valuable mindsets to the field of GNN research. For example, we fully adopt the principle of controlling model complexity when comparing different models, which is not yet adopted in most GNN papers. We focus on finding guidelines / principles when designing GNNs, rather than particular GNN instantiations. We emphasize that the best GNN designs can drastically differ across tasks (the state-of-the-art GNN model on one task may have poor performance on other tasks). We thus propose to evaluate models on diverse tasks measured by quantitative similarity metric. Rather than criticizing the weakness of existing GNN architectures, our goal is to build a framework that can help researchers understand GNN design choices when developing new models suitable for different applications. Our approach serves as a tool to demonstrate the innovation of a novel GNN model (*e.g.*, in what kind of design spaces / task spaces, a proposed algorithmic advancement is helpful), or a novel GNN task (*e.g.*, showing that the task is not similar to any existing tasks thus calls for new challenges of algorithmic development).

**Impact on machine learning research**. Our approach is in fact applicable to general machine learning model design. Specifically, we hope the proposed *controlled random search* technique can assist fair evaluation of novel algorithmic advancements. To show whether a certain algorithmic advancement is useful, it is important to sample random model-task combinations, then investigate in what scenarios the algorithmic advancement indeed improves the performance.

Additionally, the proposed *task similarity metric* can be used to understand similarities between general machine learning tasks, *e.g.*, classification of MNIST and CIFAR-10. Our ranking-based similarity metric is fully general, as long as different designs can be ranked by their performance.

**Impact on other research domains**. Our framework provides an easier than ever support for experts in other disciplines to solve their problems via GNNs. Domain experts only need to provide properly formatted domain-specific datasets, then recommended GNN designs will be automatically picked and applied to the dataset. In the fastest mode, anchor GNN models will be applied to the novel task in order to measure its similarity with known GNN tasks, where the corresponding best GNN designs have been saved. Top GNN designs in the tasks with high similarity to the novel task will be applied. If computational resources permitted, a full grid search / random search over the design space can also be easily carried out to the new task. We believe this pipeline can significantly lower the barrier for applying GNN models, thus greatly promote the application of GNNs in other research domains.

**Impact on the society**. As is discussed above, given its clarity and accessibility, we are confident that our general approach can inspire novel applications that are of high impact to the society. Additionally, its simplicity can also provide great opportunities for AI education, where students can learn from SOTA deep learning models and inspiring applications at ease.

## Acknowledgments

We thank Jonathan Gomes-Selman, Hongyu Ren, Serina Chang, and Camilo Ruiz for discussions and for providing feedback on our manuscript. We especially thank Jonathan Gomes-Selman for contributing to the `GraphGym` code repository. We also gratefully acknowledge the support of DARPA under Nos. FA865018C7880 (ASED), N660011924033 (MCS); ARO under Nos. W911NF-16-1-0342 (MURI), W911NF-16-1-0171 (DURIP); NSF under Nos. OAC-1835598 (CINES), OAC-1934578 (HDR), CCF-1918940 (Expeditions), IIS-2030477 (RAPID); Stanford Data Science Initiative, Wu Tsai Neurosciences Institute, Chan Zuckerberg Biohub, Amazon, Boeing, JPMorgan Chase, Docomo, Hitachi, JD.com, KDDI, NVIDIA, Dell. J. L. is a Chan Zuckerberg Biohub investigator. Rex Ying is also supported by Baidu Scholarship.

## Footnotes

[1]Project website with data and code: `http://snap.stanford.edu/gnn-design`

[2]`https://github.com/snap-stanford/graphgym`

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
