[Supplementary Material]

# A    Additional Implementation Details

We comprehensively discuss the GNN model specifications in Section 5, task specifications in Section 6 and key experimental setup in Section 7.2. Additional details include:

**Model**. We use L2 normalization after the final GNN layer to stabilize training, and do not use Laplacian normalization when doing message passing.

**Training**. We use cosine learning rate schedule (annealed to 0, no restarting). We use 0.0005 L2 weight decay for regularization. SGD optimizer is set with momentum of 0.9.

# B    Additional Results

In the main manuscript, we define a general design space, a general task space with quantitative similarity metric, and an effective design space evaluation technique. Our aim is *not* to cover all the design and evaluation aspects; in contrast, we wish to present a systematic framework which can inspire researchers to propose and understand new design dimensions and new tasks.

## B.1    Case Study: `Attention` as a New Design Dimension

Here we provide a case study, where we investigate `Attention` as a new intra-layer design dimension. We compare GNNs without attention, using additive attention or multiplicative attention using the same approach that we produce Figure 3 in the main manuscript. The results in Figure B.1 show that using additive attention is favorable than multiplicative attention and no attention. This is consistent with the choice of GAT where additive attention is used.

Figure B.1: **Ranking analysis for a new GNN design dimension `Attention`**

## B.2 Case Study: `Link Prediction` as a New Type of Tasks

Our GNN design framework is applicable to other graph learning tasks beyond node or graph classification tasks. Here we include additional results for link prediction tasks. Specifically, we include link prediction tasks to the task space, then re-plot Figure 1(b) in the main manuscript.

The extended task space is shown in Figure B.2. It is interesting to see that most link prediction tasks form a cluster in the task space, which is distant from node and graph classification tasks. We further show the best GNN design discovered for each Link Prediction task in Table B.1; these best designs are indeed different from best designs for node or graph classification task in Table C.1.

Figure B.2: **GNN task space for a new type of GNN tasks `Link Prediction`**

Table B.1: **Best GNN design discovered for each `Link Prediction` task**. All the performance numbers are averaged over 3 different random seeds. Task names are written as `level-dataset-feature-label`. Designs are represented as (`pre-process layer`, `message passing layer`, `post-process layer`, `layer connectivity`, `aggregation`)

| Task name | Best design in the GNN design space | Best design's performance |
|---|---|---|
| linkpred-AmazonComputers-N/A-N/A | (2, 6, 3, skipsum, mean) | 0.8185 |
| linkpred-AmazonPhoto-N/A-N/A | (1, 8, 3, skipsum, mean) | 0.8432 |
| linkpred-BZR-N/A-N/A | (2, 2, 3, skipcat, max) | 0.7394 |
| linkpred-COX2-N/A-N/A | (2, 2, 2, skipcat, max) | 0.7554 |
| linkpred-CiteSeer-N/A-N/A | (2, 2, 3, skipsum, mean) | 0.7191 |
| linkpred-CoauthorCS-N/A-N/A | (1, 8, 3, skipsum, mean) | 0.8346 |
| linkpred-CoauthorPhysics-N/A-N/A | (2, 6, 3, skipsum, mean) | 0.8273 |
| linkpred-Cora-N/A-N/A | (1, 8, 3, skipsum, mean) | 0.7305 |
| linkpred-DD-N/A-N/A | (1, 8, 2, skipcat, max) | 0.6939 |
| linkpred-ENZYMES-N/A-N/A | (1, 6, 2, skipcat, max) | 0.6253 |
| linkpred-PROTEINS-N/A-N/A | (1, 6, 2, skipsum, max) | 0.6269 |

# C   Best GNN Designs for Each Task

In Table C.1, we show the best GNN design that we discover for each task in the main manuscript. Additionally, we compare against standard GCNs with message passing layers $\{2, 4, 6, 8\}$, while using all the other optimal hyper-parameters that we have found. The best model in our design space is better than the best GCN model in 24 out of 32 tasks.

Table C.1: **Best GNN designs discovered for each task**. All the performance numbers are averaged over 3 different random seeds. Task names are written as `level-dataset-feature-label`. Designs are represented as (`pre-process layer`, `message passing layer`, `post-process layer`, `layer connectivity`, `aggregation`)

| Task name | Best design in the GNN design space | Best design's performance | Best GCN's performance |
|---|---|---|---|
| node-AmazonComputers-N/A-N/A | (1, 2, 2, skipcat, max) | 0.916 | 0.916 |
| node-AmazonPhoto-N/A-N/A | (2, 2, 2, skipcat, max) | 0.961 | 0.940 |
| node-CiteSeer-N/A-N/A | (2, 6, 2, skipcat, mean) | 0.749 | 0.714 |
| node-CoauthorCS-N/A-N/A | (1, 4, 3, skipcat, mean) | 0.952 | 0.915 |
| node-CoauthorPhysics-N/A-N/A | (1, 4, 2, skipcat, max) | 0.968 | 0.955 |
| node-Cora-N/A-N/A | (1, 8, 2, skipcat, mean) | 0.885 | 0.846 |
| node-scalefree-clustering-pagerank | (2, 8, 3, skipsum, add) | 0.977 | 0.977 |
| node-scalefree-const-clustering | (2, 8, 2, skipsum, add) | 0.712 | 0.699 |
| node-scalefree-const-pagerank | (2, 8, 2, skipsum, add) | 0.978 | 0.972 |
| node-scalefree-onehot-clustering | (1, 8, 2, skipsum, add) | 0.684 | 0.707 |
| node-scalefree-onehot-pagerank | (1, 8, 3, skipsum, add) | 0.973 | 0.973 |
| node-scalefree-pagerank-clustering | (1, 8, 2, skipsum, add) | 0.718 | 0.703 |
| node-smallworld-clustering-pagerank | (1, 8, 2, skipsum, add) | 0.957 | 0.959 |
| node-smallworld-const-clustering | (1, 8, 2, skipsum, add) | 0.607 | 0.590 |
| node-smallworld-const-pagerank | (2, 8, 3, skipsum, add) | 0.953 | 0.971 |
| node-smallworld-onehot-clustering | (1, 8, 2, skipsum, add) | 0.600 | 0.597 |
| node-smallworld-onehot-pagerank | (1, 8, 2, skipsum, add) | 0.950 | 0.967 |
| node-smallworld-pagerank-clustering | (1, 8, 2, skipsum, add) | 0.618 | 0.600 |
| graph-PROTEINS-N/A-N/A | (1, 8, 2, skipcat, mean) | 0.739 | 0.738 |
| graph-BZR-N/A-N/A | (1, 8, 2, skipcat, mean) | 0.893 | 0.894 |
| graph-COX2-N/A-N/A | (1, 6, 2, skipsum, max) | 0.809 | 0.826 |
| graph-DD-N/A-N/A | (2, 2, 3, skipsum, add) | 0.751 | 0.730 |
| graph-ENZYMES-N/A-N/A | (2, 4, 3, skipsum, add) | 0.608 | 0.504 |
| graph-IMDB-N/A-N/A | (2, 8, 2, skipsum, add) | 0.478 | 0.400 |
| graph-scalefree-clustering-path | (2, 2, 2, skipsum, add) | 0.904 | 0.814 |
| graph-scalefree-const-path | (1, 8, 3, skipcat, add) | 0.865 | 0.724 |
| graph-scalefree-onehot-path | (1, 4, 3, skipsum, add) | 0.776 | 0.686 |
| graph-scalefree-pagerank-path | (1, 8, 3, skipcat, add) | 0.840 | 0.801 |
| graph-smallworld-clustering-path | (1, 4, 2, skipcat, add) | 0.923 | 0.724 |
| graph-smallworld-const-path | (1, 8, 2, skipsum, add) | 0.865 | 0.558 |
| graph-smallworld-onehot-path | (1, 4, 3, skipcat, add) | 0.699 | 0.577 |
| graph-smallworld-pagerank-path | (1, 4, 2, skipsum, add) | 0.853 | 0.705 |
| graph-ogbg-molhiv-N/A-N/A | (2, 6, 3, skipcat, sum) | 0.792 | 0.760 |

# D   Additional Analyses

In Figure 5(b)(c) of the main manuscript, we quantitatively show that the proposed quantitative task similarity metric is a great indicator for transferring the best designs. Here we qualitatively show the high correlation by providing more visualizations. Concretely, we visualize the task similarity computed using 12 anchor models (Figure D.1), computed using all the models in the proposed design space (Figure D.2), as well as the performance ranking after task transfer (Figure D.3).

Figure D.1: **Task similarity computed using 12 anchor models**. higher value means higher similarity.

Proposed task similarity (computed from all models)

Format of tasks: Level-Dataset-Feature-Label

Figure D.2: **Task similarity computed using all the models in the design space (96 models)**

Figure D.3: **Performance ranking of the best design in one task after transferred to another task**, computed over all pairs of tasks. The ranking is normalized, higher value means better performance.