[Reviews · NeurIPS 2020]

Review 1

Summary and Contributions: This paper systemtically summarizes (1) a general GNN design space, (2) a GNN task space with quantitative similarity metric (3) an efficient and effective design space evaluation. Use this framework, they conduct GNN architecture search over a wide range of tasks, and the model discovered by them achieved sota results on large-scale benchmark.

Strengths: (1) Comprehensive and systemic summarization of existing GNN design space and tasks. (2) Get several interesting key findings that are beneficial to the communities. (3) The code is provided.

Weaknesses: (1) Some essential design choices are missing in the design space selection, such as different aggregation operations (for example, which attention mechanism to use, additive or product), pooling strategies (for example, diff pool and set pool), sampling strategies, etc. Even if the authors don't want to cover any part of these components, it would be better to discuss them and let it as future work. (2) Though the authors claim it achieved best result on the ogbn leaderboard, it would be better to add comparison with some existing standard GNN architectures. More quantitative evaluation on the proposed architecture search framework is needed. ================================================================================= Update: The authors address many of my concerns in the rebuttal, so I decide to raise up my score. But I still have some comments: It's interesting to show the comparison of attention machnism (not sure whether the product-based attention is implemented as that in Transformer, with KQV transformation). Also, I still think it would be better to also add pooling into the design space in the later version. The comparison with default architecture is very interesting, looking forward to seeing that in the later version.

Correctness: Yes, they are correct.

Clarity: Yes, the paper is well motivated and written.

Relation to Prior Work: Yes, the related work is well summarized.

Reproducibility: Yes

Additional Feedback:


Review 2

Summary and Contributions: ================ POST-REBUTTAL The rebuttal has addressed some of my concerns. The absence of theory guiding the search is really an issue not specific to this paper, but with the area. Yes, 1000+ GNN papers but only a handful of them are actually worth reading. The authors should add the “Issue of multiple hypothesis testing” remark in the final version. It will help the literature. I will be raising my score. ================ This work explores a few design options of current graph neural network architectures and reports the options that tend to reliably give better GNN architectures across most tasks.

Strengths: * The community has been proposing a few options in the design space but each paper amounts to a very small contribution. It is good to see a paper that tries to take a more systematic approach, effectively combining small contributions into a larger one. * Looking at the design decisions that give consistent improvements across architectures and tasks is a great idea. * The evaluation methodology is very good. I particularly enjoyed the violin plots and the similarity matrix between tasks. The proposed task similarity metric shows that there are only effectively three tasks in all these datasets. * It also shows that task transfer occurs on tasks that a similar but not on tasks that are different. * Significance and novelty of the contribution: The contribution is not novel. It is relevant since no other work provides a systematic evaluation.

Weaknesses: * Theoretical grounding: There is no theory guiding these design principles. There are a few tasks that test the expressiveness of neural networks. I guess all these GNNs would fail in the expressiveness tasks. * The GNN layer equation, described in the equation at the bottom of page 3, does not use the v’s own embedding. That is a strange choice, since it significantly reduces the expressiveness of the GNN. It would be good to see that as yet another hyperparameter for the model. Moreover, edge features / attention is not described by the framework, even though GAT is mentioned in the beginning of the paper. * There is no mention to the issue of multiple hypothesis testing. Anyone trying to test these many hypotheses should use a family-wise error rate correction technique to control the false discovery rate. Something simple like Bonferroni correction or Holm's step-down procedure should be described.

Correctness: The work is straightforward. There is nothing obviously wrong with the approach.

Clarity: The paper is well written.

Relation to Prior Work: The work only encompasses a small set of graph representation learning methods. Some of the more wild designs like WK-k [1], Tensor Equivariant NNs [2], and Relational Pooling [3] are not contemplated, but should be described as related works with alternative designs to message-passing GNNs. [1] Morris, C., Ritzert, M., Fey, M., Hamilton, W.L., Lenssen, J.E., Rattan, G. and Grohe, M., 2019, July. Weisfeiler and leman go neural: Higher-order graph neural networks. In Proceedings of the AAAI Conference on Artificial Intelligence (Vol. 33, pp. 4602-4609). [2] Maron, H., Ben-Hamu, H., Serviansky, H. and Lipman, Y., 2019. Provably powerful graph networks. In Advances in Neural Information Processing Systems (pp. 2156-2167). [3] Murphy, R.L., Srinivasan, B., Rao, V. and Ribeiro, B., 2019. Relational pooling for graph representations. Proceedings of the International Conference on Machine Learning (pp. 4663-4673).

Reproducibility: Yes

Additional Feedback:


Review 3

Summary and Contributions: The paper summarizes the hyper-parameters currently included in GNN, which is called design space by the author. And proposes: (1) method to find the best hyper-parameters ; (2) metrics to measure different tasks (datasets more precisely) similarity; (3) method to find optimal hyper-parameters according the tasks similarity. All of them have a certain contribution to the research community of GNN.

Strengths: (1) This paper comprehensively summarizes hyper-parameters in the current general GNN , and systematically divides them into three categories. (2) This paper proposes a method to measure tasks (datasets) similarity from an experimental point of view: to judge data-sets' similarity through the performance comparison on several anchor models. Although this method has been applied in other fields (such as NLP) for a long time, as far as I know, this paper is the first to apply this method to GNN. (3) The article does many experiments to show: (a) the overall performance of each hyper-parameter; (b) the task similarity proposed in the paper has certain guiding significance for the selection of hyper-parameters. Based on the above, this paper is a relatively complete work, and it is experiment-driven.

Weaknesses: On the whole, the biggest weakness of this paper relies on experiments too much, and lacks deep insights. The relatively new insight in this paper is: for the first time, the method of using the anchor model to judge the similarity of tasks is introduced to GNN, and the similarity is connected with the selection of hyper-parameters. However, this method of measuring similarity is experimentally driven and not elegant, nor can it accurately give the similarity. In addition, the paper gives conclusions on the effectiveness of many modules (such as dropout, BN-layer) based on experiments in section 7.3. But there is no theoretical explanation for these conclusions. For example, the GIN paper ("HOW POWERFUL ARE GRAPH NEURAL NETWORKS?") argues that injectiveness should be guaranteed as much as possible in the graph classification task, so the aggregation function chooses the sum function (this is consistent with the experimental conclusion of the author of this paper). If the author can explain these conclusions convincingly, this paper will be much more valuable.

Correctness: Yes

Clarity: Yes, the paper is presented well and easy to understand. Some small writing mistake: Line 121, it's 'inter-layer', rather than 'intra-layer'.

Relation to Prior Work: Yes

Reproducibility: Yes

Additional Feedback: --After reading author response---- This paper is experiment-driven, aiming at gaining empirical knowledge about GNN through experiments. Despite the lack of sufficient theoretical support and complete design space including attention mechanisms, the experiments designed are relatively adequate, and some interesting conclusions have been drawn. On the whole, this paper is a relatively complete submission: first, the design space is defined, then experiments are carried out and many interesting conclusions are drawn, and finally, the task similarity is defined to help new tasks quickly find the optimal hyperparameters. Furthermore, no paper has done similar work before, and this paper can promote the development of the GNN field in my opinion. Therefore, I think this paper should be published. As for the drawbacks mentioned above, I think it can be used as future work. About theoretical analysis: This paper draws many interesting conclusions. Some have already given explanations in previous work, more have not yet. The explanations for these conclusions are very valuable. About task similarity: Assuming that the metrics given in this paper are correct, it’s meaningful to introduce statistical information of the graph data (such as the distribution of degrees) to analyze which graph tasks are more similar and give convincing explanations. I have increased my score to 7.


Review 4

Summary and Contributions: The work presents a design space of GNN as while as a GNN task space. The built spaces enable a quick transfer to find the best GNN model for a new coming GNN task. === POST REBUTTAL REVIEW Thanks for the author's effort in the rebuttal. The additional experiment result looks cool. I increase my score to 8.

Strengths: The paper is novel. Instead of optimizing GNN for one specific task. It takes the step back and trying to build a space for the GNN task and looking for a systematic way to discover best GNN model for all tasks. Good perspective. The technique is sound. The way it uses ranking to evaluate the model’s performance and the design of task similarity based on the anchor model’s performance are reasonable and empirically validated. The paper also provides a code platform, UBIGRAPH, which facilitates the GNN model design for the practitioners. It is very significant. Finally, the paper is clearly written.

Weaknesses: === diversity of the space === The GNN design space proposed in the paper is still in the range of the common GNN structure. It limits the system to “invent” GNN which beyond handcrafted GNN designs. Also, some existing GNN structures such as attention[0] are not included in the current design space. Similarly, the diversity issue also exists in the task space. Currently, the task space includes 32 tasks but as the later analysis shows they mainly lie in two groups. For example, one common GNN task, link prediction is not considered in this task space. I totally understand building a system to including everything is hard. I guess the current system already requires a significant amount of effort. Appreciate that. == comparison to the NAS === In the case study, the paper shows the system can be leverage to find a good GNN model a new task. I am curious about the result of applying NAS (such as auto-GNN[1]) on this ogbg-moihiv dataset. I would like to see the difference between the performance as while as the computation power and model search time required. Including such comparison would make us understand the pros and cons better about the system. [0] graph attention networks [1] Auto-GNN: Neural Architecture Search of Graph Neural Networks

Correctness: yes

Clarity: yes

Relation to Prior Work: yes

Reproducibility: Yes

Additional Feedback:

[Author Response · NeurIPS 2020]

We thank the reviewers for their constructive feedback. All reviewers point out that our paper presents *the first* systematic

approach to study GNN designs, *the first* quantitative analysis for GNN task similarity, and offers rigorous findings via

novel evaluation techniques. With 1000+ new GNN papers each year, we hope our framework can greatly facilitate the

design and evaluation of GNNs. Reviewers ask for clarifications and new experiments, which we answer below:

**1 Lack of theoretical analysis (R2 R3)** We thank R2 and R3 for raising that our paper lacks theoretical analysis.

Indeed, our paper focuses on empirical understandings of GNN design: the novelty of our systematic framework and

valuable findings are acknowledged by all the reviewers. Here we emphasize that our framework provides a solid tool

that can *verify and inspire theoretical findings*. For instance, the GIN paper shows the nice theoretical result that SUM

aggregation is more expressive than MEAN and MAX; however, their evaluation can be improved, *e.g.*, only compare on

a fixed GNN design (5-layer, 64-dim, etc.) on a few graph classification tasks. In contrast, our framework samples

hundreds of models from 10M possible model-task combinations, *with every design dimensions controlled except the*

*aggregation function*, which is the first comprehensive and rigorous evaluation that verifies SUM is indeed empirically

successful (Fig 3). Similarly, our framework provides *rigorous evidence* to other theoretical results *in the context*

*of GNN*, *e.g.*, BN helps neural network training, skip connections avoid the problem of vanishing gradients. More

interestingly, our paper makes the novel discovery that PRELU activation significantly improves GNN performance.

We think this finding suggests the uniqueness of GNN optimization landscape, and hope it can inspire theoretical works

towards the open question of improving GNN optimization. We will add these new discussions to the revised paper.

**2 Additional design dimensions (R1 R2 R4).** We thank reviewers for suggesting other design dimensions to explore.

We defined a general design space including intra-layer design, inter-layer

design and learning configurations; however, we were not able to cover

all aspects, and especially thank R4's appreciation for our efforts. We

wish to present a systematic framework which can inspire researchers to

propose and understand new design dimensions – reviewers' constructive

suggestions in fact illustrate the importance of such a framework. Based

on these suggestions, we run new experiments. **New results for attention**

**(R1 R2 R4).** We compare GNNs without attention, using additive attention

or multiplicative attention using the same approach that we produce Fig 3.

The results show that using additive attention is favorable than multiplica-

tive attention and no attention. This is consistent with the choice of GAT

where additive attention is used. **New results for link prediction (R4).**

Following R4's suggestion, we additionally include link prediction tasks on

Cora and ENZYMES to the task space. The best architecture we found for

Cora is "(1, 8, 3, skipsum, mean, 400)", for ENZYMES is "(1, 6, 2, skipcat,

max, 400)" (*c.f.*, Fig 1(c) in our paper). Interestingly, by visualizing the task

embeddings via the proposed task similarity metric, we find link prediction

on Cora is different from other tasks, while link prediction on ENZYMES

is similar to some node classification tasks. We will include these new results in the revised version.

**3 More comparisons 1) With standard architectures (R1).** We thank R1 for asking the comparison with standard

GNN architectures. We emphasize that the goal of our paper is not pursuing STOA performance, but presenting a

systematic approach for GNN design. In fact, our systematic approach can be used to determine the hyperparameters

of existing architectures. Following R1's suggestion, we implement standard GCNs with message passing layers

$\{2, 4, 6, 8\}$, while *keeping all the other optimal hyper-parameters we discovered* in line 268. The best model in our

design space is better than the best GCN model in 24 out of 32 tasks. Note that we defined a simple GNN design space.

Our new results show that adding attention further improve the performance. We will include these new results. **2) With**

**NAS approaches (R4).** Our framework is orthogonal to NAS approach: we focus on designing and evaluating a search

space, while NAS approaches focus on finding the best model from a given search space. Unfortunately, applying

Auto-GNN on the large ogbg-molhiv dataset requires training 2000+ models which is beyond our computing resources.

**4 Related work (R2).** We thank R2 for pointing out other powerful GNNs and will cite them in the revised version.

**5 Clarifications**. **Q(R2):** "Issue of multiple hypothesis testing" **A:** We thank R2 for pointing out the issue. We resample

experiments for each design dimension in Fig 3 so this is less of concern. Nevertheless, we run one-way ANOVA with

Bonferroni correction (p-value 0.05). 8 (without correction) and 7 (after correction) out of the 12 design dimensions

have significant findings. **Q(R2):** "Use $v$'s own embedding in message passing" **A:** The SKIP-SUM design choice that

we use is equivalent to what R2 suggests. **Q(R3):** "Experiment-driven task similarity" **A:** We agree with R3 that our

approach can be improved; however, how to define the "real" similarity between tasks is still an open question. We are

the first who introduce the notion of task similarity to the GNN community, and we provide strong evidence that the

proposed task similarity is useful (Fig 5). **Q(R2 R4):** "In the range of common GNNs" **A:** R2 and R4 are correct that

the models we consider are common GNNs, thus will fail in expressiveness tasks. Designing more powerful GNNs is

still an open domain which cannot be summarized into a design space yet, therefore we do not include in our paper.

[Meta-Review · NeurIPS 2020]

The paper systematically studies neural architecture search for graph neural networks by proposing (1) a general GNN design space, (2) a GNN task space with a quantitative similarity metric and (3) the design space evaluation. Although it is experiment-driven and lacks deep insights or theoretical analysis, the comprehensive and systemic evaluation of GNN design are important to the community. Based on the reviews, the merits of the paper outweigh the drawbacks and acceptance is recommended.